# Protection-Free Strategy for the Synthesis of Boro-Depsipeptides in Aqueous Media under Microwave-Assisted Conditions

**DOI:** 10.3390/molecules27072325

**Published:** 2022-04-04

**Authors:** Shuo-Bei Qiu, Pei-Yao Liu, Bo-Cheng Wang, Pin-Rui Chen, Jing-Han Xiao, Ting-Yu Hsu, Kuan-Lin Pan, Zhi-Yin Lai, Yi-Wei Chen, Ying-Chuan Chen, Jen-Kun Chen, Po-Shen Pan

**Affiliations:** 1Graduate Institute of Life Science, National Defense Medical Center, No. 161, Sec. 6, Minquan E. Rd., Neihu Dist., Taipei City 11490, Taiwan, Republic of China; gloria919147@gmail.com; 2Department of Physiology & Biophysics, National Defense Medical Center, No. 161, Sec. 6, Minquan E. Rd., Neihu Dist., Taipei City 11490, Taiwan, Republic of China; peiyaol01@gmail.com; 3Department of Chemistry, College of Sciences, Tamkang University, No. 151, Yingzhuan Rd., Danshui Dist., New Taipei City 25137, Taiwan, Republic of China; bcw@mail.tku.edu.tw (B.-C.W.); shawn19991002@gmail.com (P.-R.C.); a0952389829@gmail.com (J.-H.X.); lydia1020552@gmail.com (T.-Y.H.); ss870119@gmail.com (K.-L.P.); aa0935900324@gmail.com (Z.-Y.L.); 4Department of Oncology, Taipei Veterans General Hospital, No. 201, Sec. 2, Shipai Rd., Beitou District, Taipei City 11217, Taiwan, Republic of China; chenyw@vghtpe.gov.tw; 5Center for Nanomedicine Research, National Health Research Institutes, Zhunan, Miaoli County 35053, Taiwan, Republic of China

**Keywords:** depsipeptide(s), α-acyloxyamide, passerini, multicomponent, microwave-assisted, DAHMI, boronic acid

## Abstract

In this report, 19 boron-containing depsipeptides were synthesized via microwave-assisted Passerini three-component reaction (P-3CR) in an aqueous environment. The linker-free DAHMI fluorescent tagging approach was used on selected boron-containing compounds to study the relationship between their structures and their level of cellular uptake of HEK293 cells. The biological data retrieved from the DAHMI experiments indicated that while the structures of tested compounds may be highly similar, their bio-distribution profile could be vastly distinctive. The reported optimized one-pot synthetic strategy along the linker-free in vitro testing protocol could provide an efficient platform to accelerate the development of boron-containing drugs.

## 1. Introduction

Boronic acids are important entities for a broad spectrum of applications [1]. Although they were widely recognized as the vital building blocks in varieties of synthetic reactions [2], boron-containing entities were also exploited as the flame retardant materials [3], semi-conducting materials [4], boron carrier agents for the boron neutron capture therapy [5], enzyme inhibitors [6], and also as the insecticide [7]. In particular, boron-based pharmaceuticals have gained much attention and success for the past decades. Unlike most pharmaceuticals that rely on non-covalent interactions with their biological targets, boron-containing compounds form a reversible covalent bond between their targets. This unique advantage makes such compounds have a high affinity toward their targets with relatively low molecular weight [6]. In 2003, the first boronic acid, Bortezomib (Velcade^®^), was approved by the U.S. FDA to treat multiple myeloma. Since then, a series of boron-containing molecules are either in different stages of clinical trials or have been approved by the U.S. FDA (Figure 1).

Due to the presence of an empty *p*-orbital, boronic acids are exceptional candidates for inhibiting hydrolytic enzymes, which are often overexpressed in ranges of diseases [8]. The strategies involved in the making of boronic acid analogues often demand the protection of the boronic acid group prior to any synthetic operations. However, removing the protecting group often requires a harsh condition or extended reaction period, making this process remarkably unfavorable. In addition, the purification of boron-containing substrates after each synthetic step has proven challenging and is considered an intense labor operation [9]. In this regard, the direct synthesis of boronic acids using unprotected boron building blocks via a one-pot multicomponent reaction is highly desirable [10].

One of the most realized multicomponent reactions, Passerini three-component reaction (P-3CR) has been extensively used in medicinal programs to generate depsipeptide analogues. It has successfully led to the discovery of numerous U.S. FDA approval drugs such as Diltiazem (Cardizem^TM^), Trimethadione (Tridione^TM^), Paramethadione (Paradione^TM^), Camazepam (Albego^TM^), Roxatidine (Roxit^TM^) and Fusafungine (Locabiotal^TM^). In 2018, Eduardo Peña-Cabrera and co-workers published their work on using formyl aryl boronic acids as one of the building blocks to construct a series of Passerini analogues [11]. In the following year, Alexander Dömling and co-workers reported that a series of free phenylboronic acids were utilized as building blocks in numerous IMCR platforms, including P-3CR [12]. Both reports have demonstrated that free boronic acids could be utilized directly in IMCRs. Inspired by their work, we developed microwave-assisted conditions that allow the rapid assembly of a series of P-3CR analogues by unprotected phenylformyl/phenylcarboxyl boronic acids. The reactions are environmentally benign as they proceed in H_2_O, and the crudes could be simply purified by acid/base extractions and re-crystallization.

## 2. Results and Discussion

### 2.1. Chemistry

In this report, two series of boron-containing P-3CR analogues were synthesized. Synthetic conditions for each series were optimized correspondingly and are summarized in Table 1 and Table 2. The first series of compounds were synthesized using boron-containing aldehyde as one of the building blocks, and the optimization processes are summarized in Table 1. Four commonly used polar solvents were utilized for the direct synthesis of depsipeptide boronic acid **4a** (Table 1, entry 1–4). It was found that under room temperature, only H_2_O gave the desired product in 2.5 h (Table 1, entry 4). Increasing the reaction temperature from room temperature to 45 °C with H_2_O significantly improved the yield from 5% to 56% (Table 1, entry 6). It is important to note that, at 45 °C, the one with microwave irradiation (Table 1, entry 8, 68%) gave a higher yield than the one with the oil bath (Table 1, entry 6, 56%). This could be the consequence of the more efficient energy input by the microwave heating condition. Higher temperatures were also tested (Table 1, entry 10–11); however, the yields were either similar (65 °C) or decreased (85 °C). Reducing the reaction time from 2.5 h to 1 h gave a lower yield (Table 1, entry 7, 55%), indicating the reaction has yet to be completed. However, increasing the reaction time to 4 h (Table 1, entry 9, 33%) does not benefit the yield. Presumably, the side reactions were taking place where we observed the increased number of unidentified signals from the crude ^1^H NMR spectrum. Hence the optimal condition was to react in H_2_O under microwave-assisted heating at 45 °C for 2.5 h (Table 1, entry 8).

With the optimized condition in hand, seven of the additional compounds were synthesized (**4a**–**g**), and the results were summarized in Figure 1.

Initially, the optimal condition retrieved from Table 1 was applied to construct second series of analogues, in which the boron-containing benzoic acids were used as one of the building blocks (Table 2, entry 4). It was found that the yield would be significantly improved if the reaction temperature was raised from ambient temperature to 85 °C under microwave-irradiation condition (Table 2, entry 1, 4–6). However, a significant decline in yield was observed when the temperature reached 95 °C. Presumably, because the boiling point of the *t*-butyl isocyanide is 91 °C, rendering the reaction proceeded without the presence of the sufficient *t*-butyl isocyanide (Table 2, entry 7). It is noteworthy that at 85 °C, the one with microwave irradiation (Table 2, entry 6, 89%) gave a higher yield than the one with the oil bath (Table 2, entry 3, 53%). This should also contribute to the more efficient energy input by the microwave heating condition.

With the optimized condition in hand, the second series of P-3CR analogues (**5a**–**l**) were prepared (Figure 2). 

### 2.2. Theoretical Calculations

Although the desired compounds were constructed via the Passerini reaction, we noticed that simply replacing a boron-containing aldehyde with its carboxylic acid counterpart would considerably improve the yield from 8% (Figure 1, **4f**) to 21% (Figure 2, **5d**) under the same synthetic condition. This result indicated that the location of the boronic acid group might play a critical role in determining the outcome of the P-3CR. A series of theoretical calculations and spectroscopic studies were executed to further investigate how the boron group engages the reaction mechanism. 

Due to an empty *p*-orbital, boron is prone to nucleophilic attack [13]. Therefore, we hypothesized that adding the building block with the boronic acid group in P-3CR could alter the overall reaction mechanism. For instance, when an isocyanide approaches the boron-containing protonated aldehyde substrate (PAS), it could either attack the boronic acid group (Figure 2a, pathway A) or the protonated aldehyde group (Figure 2a, pathway B). Likewise, when the boron-containing carboxylate substrate (CS) was used, the isocyanide could attack not only the nitrilium intermediate but also the boronic acid group (Figure 2b, pathway C) or the carboxylate group (Figure 2b, pathway D).

To corroborate how isocyanide engaged between the protonated aldehyde group or the carboxylate group, a series of theoretical calculations were executed using a DFT/B3LYP/6-31G(d,p) method. It was found that when the protonated aldehyde substrate (PAS) was presented in the phenylboronic acid system (Table 3, entry 1, LUMO-side/top), the electron density of the phenyl group did not delocalize to the boron’s empty *p*-orbital. In other words, it increases the possibility of isocyanide attacking the boronic acid group. On the other hand, when the phenylboronic acid was attached to the carboxylate substrate (CS), the electron density of the phenyl group delocalized to the boron’s empty p-orbital, which protected it from being attacked by the isocyanide (Table 3, entry 2, LUMO-side/top).

Projected density of state (pDOS) analysis of the HOMO and LUMO states of boron-containing PAS and CS systems was also obtained using the DFT/B3LYP/6-31G(d,p) method, and the results are summarized in Table 4. It was found that when the protonated formyl group is presented, the relative charge density of boron was 0.73 in the LUMO state (Table 4, entry 1); however, when the carboxylate group is presented, it increased drastically to 19.85 (Table 4, entry 2). These results implied that when the electron-withdrawing protonated aldehyde group is presented, it is prone to nucleophilic attack from the isocyanide. On the other hand, when an electron-donating carboxylate is presented, the delocalization of the electrons from the phenyl group protects the boronic acid from such an attack.

Two model reactions were carried out in the NMR tubes to validate the abovementioned theoretical calculations, and their corresponding ^11^B NMR spectrums are shown below (Figure 3).

The in-situ ^11^B NMR studies of P-3CR have shown that when a boron-containing aldehyde is used (Figure 3a, Model A), a signal representing the formation of tetra-coordinate boron at 18.66 ppm emerged right next to a tri-coordinate boronic acid signal (27.18 ppm). On the other hand, when a boron-containing carboxylic acid was used (Figure 3b, Model B), the tetra-coordinate boron signal was almost unnoticeable. The results were in line with the theoretical calculations and explained how a boron-containing building block impacted the overall results of the multicomponent reaction. To the best of our knowledge, this is the first report discussing the involvement of boronic acid in the isocyanide-based multicomponent reaction.

### 2.3. In Vitro Studies

One of the critical aspects of developing a pharmaceutical agent is to elucidate its bio-distribution profile fully. The study often involves placing the fluorescent tag onto the analogs via a suitable linker. The whole process involves labor-intense efforts and is often considered as the bottleneck of the drug discovery program.

Boron Neutron Capture Therapy (BNCT) is a binary biochemically targeted radiotherapy where the boron carrier agent and thermal neutron are both required to execute the treatment procedure. Compared to conventional radiation therapy and chemotherapy, BNCT can destroy the tumor cells while leaving normal tissues unharmed. One of the gold standards required for the ideal boron delivery agent is low toxicity and high tumor uptake [14,15].

In 2018, Eduardo Peña-Cabrera and co-workers reported a labeling strategy where a novel fluorescent tag was attached to the boronic acids via Liebeskind-Srogl cross-coupling prior to the biological evaluations. Their approach provides an outstanding alternative for cell imaging [11].

In this study, a linker-free tagging strategy specifically for the boronic acid analogs was applied. Developed by Kirihata and co-workers [16], this strategy utilizes 5-(diethylamino)-2-((methylimino)methyl)phenol (DAHMI) that can complex with the boronic acid group to show fluorescent activity in cells. Compared to the conventional boron detection approaches such as α-autoradiography [17] and immune-staining [18], this strategy offers a much more appealing option as it requires no pre-installation of the linker and could be reliably deployed under aqueous conditions.

All compounds except **4e**, **5a**, **5h**, **5i**, **5l** (for their poor solubility issues) were subjected to the cytotoxicity experiments using the HEK293 cell line. The compounds tested are all possessed of IC_50_ values above 50 μM (Appendix A), which is promising for use in BNCT as the boron carrier agent. Compounds **4a**, **5b**, **5f**, **5g**, and **5k** were selected for the DAHMI experiments to illustrate the importance of estimating the biodistribution profiles by the similarity of the structures between compounds, and the results were summarized in Table 5. Although it was expected to see compounds with distinctive structures such as **4a** and **5b** to have utterly different uptake patterns, we were surprised to find that even structurally similar compounds such as **5f** and **5g** also exhibited noticeable differences. Furthermore, such findings could also be found between **5g** and **5k**, in which two comparable analogs showed a completely different level of cellular uptake. Thus, contradicting the conventional aspects, the results indicated that similar analogues do not necessarily share comparable cellular bio-distribution profiles.

## 3. Materials and Methods

### 3.1. Chemistry

#### 3.1.1. General Information

All starting materials were obtained from commercial suppliers and used without further purification unless otherwise noted. Reactions were performed in a CEM Discover Benchmate™ microwave reactor with sealed vessels.

#### 3.1.2. Spectroscopic Measurements

Unless otherwise specified ^1^H, ^13^C NMR and ^11^B NMR spectra were recorded on a Bruker Avance 600 FT-NMR spectrometer at 192.5 MHz. Data are represented as follows: chemical shifts (ppm), multiplicity (s: singlet, d: doublet, t: triplet, m: multiplet, br: broad), coupling constant *J* (Hz). High-resolution ESI mass spectra were performed on Waters LCT Premier XE (Waters Corp., Manchester, UK).

#### 3.1.3. General Procedure for Compound **4a**–**g**

Carboxylic acid (1.2 equiv.), aldehyde (1.0 equiv.) and deionised H_2_O (1.0 M) were treated for 6 min under microwave irradiation (45 °C, 150 W). Isocyanide (1.2 equiv.) was then added to the reaction mixture. Additional microwave irradiation was applied for 2.5 h (45 °C, 150 W) under medium speed magnetic stirring. After the reaction was completed, the crude material was concentrated and re-dissolved in dichloromethane. The resulting organic solution was then washed with 1 M HCl (aq). This was followed by adding a saturated aqueous solution of NaHCO_3_ (aq) combined with brine. The resulting organic layer was collected, dried by MgSO_4_, and filtered. The filtrate was then concentrated in vacuo and the crude was precipitated by dichloromethane/H_2_O or dichloromethane/hexane sonication, and the resulting solid was collected by filtration to give the desire product **4a**–**g**.

#### 3.1.4. Synthesis and Characterization of Compound **4a**–**g**

##### (4-(2-(Tert-Butylamino)-1-(Isobutyryloxy)-2-Oxoethyl)phenyl)boronic Acid (**4a**)

Following general procedures, the desired compound was synthesized utilizing isobutyric acid (0.15 mL, 1.60 mmol), 4-formylphenylboronic acid (200 mg, 1.30 mmol), and tert-butyl isocyanide (0.18 mL, 1.60 mmol), giving compound **4a** as a yellow oil (yield 283.9 mg, 68%) ^1^H NMR (600 MHz, CD_3_OD) δ 7.77 (br, 2H), 7.47 (d, *J* = 7.8 Hz, 2H), 5.86 (s, 1H), 2.71–2.67 (m, 1H), 1.29 (s, 9H), 1.20 (d, *J* = 7.2 Hz, 3H), 1.16 (d, *J* = 7.2 Hz, 3H). ^13^C NMR (151 MHz, CD_3_OD) δ 177.6, 170.2, 138.9, 135.1, 127.5, 76.8, 52.4, 35.0, 28.9, 19.4, 19.3. ^11^B NMR (192.5 MHz, CD_3_OD) δ 28.5. HRMS (ESI^+^): *m*/*z*: Calcd. for C_16_H_25_BNO_5_ [M + H]^+^: 322.1820; Found: 322.1808. 

##### (3-(2-(Tert-Butylamino)-1-(Isobutyryloxy)-2-Oxoethyl)phenyl)boronic Acid (**4b**)

Following the general procedures, the desired compound was synthesized utilizing isobutyric acid (0.07 mL, 0.80 mmol), 3-formylphenylboronic acid (100 mg, 0.66 mmol), and tert-butyl isocyanide (0.09 mL, 0.80 mmol), giving compound **4b** as a yellow oil (yield 31.8 mg, 15%). ^1^H NMR (600 MHz, CD_3_OD) δ 7.88 (br, 1H), 7.75 (br, 1H), 7.52 (d, *J* = 7.8 Hz, 1H), 7.34 (t, *J* = 7.2 Hz, 1H), 5.86 (s, 1H), 2.72–2.67 (m, 1H), 1.30 (s, 9H), 1.21 (d, *J* = 7.2 Hz, 3H), 1.17 (d, *J* = 7.2 Hz, 3H). ^13^C NMR (151 MHz, CD_3_OD) δ 177.7, 170.5, 136.3, 135.2, 134.1, 130.2, 128.8, 77.0, 52.3, 34.9, 28.8, 19.3, 19.2. ^11^B NMR (192.5 MHz, CD_3_OD) δ 28.1. HRMS (ESI^+^): *m*/*z*: Calcd. for C_16_H_25_BNO_5_ [M + H]^+^: 322.1820; Found: 322.1811. 

##### (4-(2-(Tert-Butylamino)-1-(Isobutyryloxy)-2-Oxoethyl)-2-Fluorophenyl)boronic Acid (**4c**)

Following general procedures, the desired compound was synthesized utilizing isobutyric acid (0.06 mL, 0.71 mmol), 2-fluoro-4-lphenylboronic acid (100 mg, 0.60 mmol), and tert-butyl isocyanide (0.08 mL, 0.71 mmol), giving compound **4c** as a yellow oil (yield 91.6 mg, 45%). ^1^H NMR (600 MHz, CD_3_OD) δ 7.73 (s,1H), 7.29 (d, *J* = 7.8 Hz, 1H), 7.19 (d, *J* = 10.2 Hz, 1H), 5.84 (s, 1H), 2.73–2.68 (m, 1H), 1.29 (s, 9H), 1.21 (d, *J* = 7.2 Hz, 3H), 1.18 (d, *J* = 7.2 Hz, 3H). ^13^C NMR (151 MHz, CD_3_OD) δ 177.5, 169.7, 167.5 (d, *J* = 258.2 Hz, CF) 141.4, 136.7, 123.6, 122.1, 114.7, 114.5, 76.0, 52.4, 34.9, 28.8, 19.3, 19.2. ^11^B NMR (192.5 MHz, CD_3_OD) δ 28.3. HRMS (ESI^+^): *m*/*z*: Calcd. for C_16_H_24_BFNO_5_ [M + H]^+^: 340.1726; Found: 340.1717.

##### (3-(2-(Tert-Butylamino)-1-(Isobutyryloxy)-2-Oxoethyl)-4-Methoxyphenyl)boronic Acid (**4d**)

Following general procedures, the desired compound was synthesized utilizing isobutyric acid (0.06 mL, 0.71 mmol), 3-formyl-4-methoxyphenylboronic acid (100 mg, 0.60 mmol), and tert-butyl isocyanide (0.08 mL, 0.71 mmol), giving compound **4d** as a yellow oil (yield 208.6 mg, 99%). ^1^H NMR (600 MHz, CD_3_OD) δ 7.82 (s, 1H), 7.77 (d, *J* = 8.4 Hz, 1H), 6.98 (t, *J* = 8.4 Hz, 1H), 6.21 (s, 1H), 3.87 (s, 3H), 2.17–2.66 (m, 1H), 1.33 (s, 9H), 1.20 (d, *J* = 6.6 Hz, 3H), 1.16 (d, *J* = 6.6 Hz, 3H). ^13^C NMR (151 MHz, CD_3_OD) δ 177.7, 170.5, 160.2, 137.4, 135.5, 124.4, 111.1, 71.9, 56.0, 52.3, 35.0, 28.8, 19.4, 19.2. ^11^B NMR (192.5 MHz, CD_3_OD) δ 28.3. HRMS (ESI^+^): *m*/*z*: Calcd. for C_17_H_26_BNO_6_ [M + H]^+^: 352.1925; Found: 352.1918.

##### (5-(2-(Tert-Butylamino)-1-(Isobutyryloxy)-2-Oxoethyl)-2-Fluorophenyl)boronic Acid (**4e**)

Following general procedures, the desired compound was synthesized utilizing isobutyric acid (0.07 mL, 0.72 mmol), 2-fluoro-4-formylphenylboronic acid (100 mg, 0.60 mmol), and tert-butyl isocyanide (0.08 mL, 0.72 mmol), giving compound **4e** as a yellow oil (yield 134.3 mg, 66%). ^1^H NMR (600 MHz, CD_3_OD) δ 7.72 (br, 1H), 7.55–7.53 (m, 1H), 7.06 (t, *J* = 9.0 Hz, 1H), 5.83 (s, 1H), 2.72–2.67 (m, 1H), 1.29 (s, 9H), 1.20 (d, *J* = 6.6 Hz, 3H), 1.16 (d, *J* = 6.6 Hz, 3H). ^13^C NMR (151 MHz, CD_3_OD) δ 177.6., 167.7 (d, *J* = 241.6 Hz, CF), 135.8, 132.7, 132.3, 116.2, 116.1, 76.2, 52.4, 34.9, 28.8, 19.3, 19.2. ^11^B NMR (192.5 MHz, CD_3_OD) δ 28.6. HRMS (ESI^+^): *m*/*z*: Calcd. for C_16_H_23_BFNO_5_ [M + Na]^+^: 362.1546; Found: 362.1546.

##### (4-(1-(Benzoyloxy)-2-(Tert-Butylamino)-2-Oxoethyl)phenyl)boronic Acid (**4f**)

Following general procedures,, the desired compound was synthesized utilizing benzoic acid (0.12 mL, 0.80 mmol), 4-formyl phenylboronic acid (100 mg, 0.67 mmol), and tert-butyl isocyanide (0.10 mL, 0.80 mmol) giving compound **4f** as a yellow oil (yield 101.1 mg, 45%). ^1^H NMR (600 MHz, CD_3_OD) δ 8.08 (d, *J* = 7.8 Hz, 2H), 7.87 (s, 1H), 7.79 (s, 1H), 7.58 (q, *J* = 22.2 Hz, 3H), 7.46 (t, *J* = 15.6 Hz, 2H), 6.08 (s, 1H), 1.30 (s, 1H). ^13^C NMR (151 MHz, CD_3_OD) δ 170.4, 167.2, 135.3, 134.9, 134.7, 130.9, 129.7, 127.6, 77.5, 52.6, 28.9. ^11^B NMR (192.5 MHz, CD_3_OD) δ 28.6. HRMS (ESI^+^): *m*/*z*: Calcd. for C_19_H_23_BNO_5_ [M + H]^+^: 356.1663; Found: 356.1656.

##### (4-(2-(Tert-Butylamino)-2-oxo-1-((pyrazine-2-Carbonyl)oxy)ethyl)phenyl)boronic Acid (**4g**)

Following general procedures, the desired compound was synthesized utilizing pyrazine-2-carboxylic acid (99.30 mg, 0.80 mmol), 4-formyl phenylboronic acid (100 mg, 0.67 mmol), and tert-butyl isocyanide (0.10 mL, 0.80 mmol), giving compound **4g** as a yellow oil (yield 160.3 mg, 67%). ^1^H NMR (300 MHz, CD_3_OD) δ 9.32 (s, 1H), 8.84 (d, *J* = 2.7 Hz, 1H), 8.76 (s, 1H), 7.79–7.75 (m, 1H), 7.58 (s, 2H), 7.39 (d, *J* = 13.8 Hz, 1H), 6.16 (s, 1H), 1.33 (s, 9H). ^13^C NMR (151 MHz, CD_3_OD) δ 174.4, 169.8, 164.4, 149.6, 147.5, 146.1, 144.3, 135.1, 127.9, 127.2, 78.5, 75.7, 29.0. ^11^B NMR (192.5 MHz, CD_3_OD) δ 28.4. HRMS (ESI^+^): *m*/*z*: Calcd. for C_17_H_20_BN_3_O_5_ [M + Na]^+^: 380.1388; Found: 380.1385.

#### 3.1.5. General Procedure for Compound **5a**–**l**

Carboxylic acid (1.2 equiv.), aldehyde (1.0 equiv.), and deionised H_2_O (1.0 M) were reacted for 6 min under microwave irradiation (85 °C, 150 W). Isocyanide (1.2 equiv.) was then added to the reaction mixture. The additional microwave irradiation was applied for 2.5 h (85 °C, 150 W) under medium speed magnetic stirring. After the reaction was completed, the crude material was concentrated and re-dissolved in dichloromethane. The resulting organic solution was then washed with 1 M HCl (aq). This was followed by adding a saturated aqueous solution of NaHCO_3_ (aq) combined with brine. The resulting organic layer was collected, dried by MgSO_4_, and filtered. The filtrate was then concentrated in vacuo and the crude was precipitated by dichloromethane/H_2_O or dichloromethane/hexane sonication, and the resulting solid was collected by filtration to afford the desired product **5a**–**l**. 

#### 3.1.6. Synthesis and Characterization of Compound **5a**–**l**

##### (4-((2-(Tert-Butylamino)-2-Oxoethoxy)carbonyl)phenyl)boronic Acid (**5a**)

Following general procedures, the desired compound was synthesized utilizing 4-boronobenzoic acid (331.55 mg, 2.00 mmol), formaldehyde (0.16 mL, 1.67 mmol), and tert-butyl isocyanide (0.23 mL, 2.00 mmol), giving compound **5a** as a yellow oil (yield 344.9 mg, 74%).^1^H NMR (600 MHz, CD_3_OD) δ 8.02 (d, *J* = 7.2 Hz, 2H), 7.86 (br, 2H), 4.69 (s, 2H), 1.36 (s, 9H). ^13^C NMR (151 MHz, CD_3_OD) δ 168.8, 167.5, 134.9, 129.7, 64.2, 52.2, 28.9. ^11^B NMR (192.5 MHz, CD_3_OD) δ 28.5. HRMS (ESI^+^): *m*/*z*: Calcd. for C_13_H_19_BNO_5_ [M + H]^+^: 280.1350; Found: 280.1341. 

##### (3-((2-(Tert-Butylamino)-2-Oxoethoxy)carbonyl)phenyl)boronic Acid (**5b**)

Following general procedures, the desired compound was synthesized utilizing 3-boronobenzoic acid (331.55 mg, 2.00 mmol), formaldehyde (0.16 mL, 1.67 mmol), and tert-butyl isocyanide (0.23 mL, 2.00 mmol), giving compound **5b** as a yellow oil (yield 256.4 mg, 55%). ^1^H NMR (600 MHz, CD_3_OD) δ 8.45 (br, 1H), 8.08 (d, *J* = 7.8 Hz, 1H), 8.00 (br, 1H), 7.48 (t, *J* = 7.6 Hz, 1H), 6.04 (br, 1H), 4.71 (s, 2H), 1.39 (s, 9H). ^13^C NMR (151 MHz, CD_3_OD) δ 166.6, 165.5, 138.9, 134.9, 131.6, 128.3, 128.1, 63.5, 51.6, 28.7. ^11^B NMR (192.5 MHz, CD_3_OD) δ 29.0. HRMS (ESI^+^): *m*/*z*: Calcd. for C_13_H_19_BNO_5_ [M + H]^+^: 280.1350; Found: 280.1347. 

##### (4-((2-(Tert-Butylamino)-2-Oxoethoxy)carbonyl)-3-Fluorophenyl)boronic Acid (**5c**)

Following general procedures, the desired compound was synthesized utilizing 4-borono-2-fluorobenzoic acid (367.49 mg, 2.00 mmol), formaldehyde (0.16 mL, 1.67 mmol), and tert-butyl isocyanide (0.23 mL, 2.00 mmol), giving compound **5c** as a yellow oil (yield 284.3 mg, 61%).^1^H NMR (600 MHz, CD_3_OD) δ 7.90 (t, *J* = 7.8 Hz, 1H), 7.58 (br, 1H), 7.48 (br, 1H), 4.67 (s, 2H), 1.36 (s, 9H). ^13^C NMR (151 MHz, CD_3_OD) δ 168.5, 164.8, 162.7 (d, *J* = 259.7 Hz, CF), 132.3, 130.4, 122.7, 122.6, 119.9, 64.3, 52.4, 29.0. ^11^B NMR (192.5 MHz, CD_3_OD) δ 27.7. HRMS (ESI^+^): *m*/*z*: Calcd. for C_13_H_17_BFNO_5_ [M + H]^+^: 298.1256; Found: 298.1245.

##### (4-((2-(Tert-Butylamino)-2-Oxo-1-Phenylethoxy)carbonyl)phenyl)boronic Acid (**5d**)

Following general procedures, the desired compound was synthesized utilizing 4-boronobenzoic acid (93.83 mg, 0.57 mmol), benzaldehyde (0.05 mL, 0.47 mmol), and tert-butyl isocyanide (0.06 mL, 0.57 mmol), giving compound **5d** as a yellow oil (yield 75.1 mg, 45%). ^1^H NMR (600 MHz, CD_3_OD) δ 8.04 (d, *J* = 7.8 Hz, 2H), 7.85 (br, 1H), 7.59 (d, *J* = 7.8 Hz, 2H), 7.37 (m, 3H), 6.07 (s, 1H), 1.30 (s, 9H). ^13^C NMR (151 MHz, CD_3_OD) δ 170.4, 167.3, 140.7, 137.3, 134.9, 131.4, 130.9, 130.0, 77.5, 52.5, 28.9. ^11^B NMR (192.5 MHz, CD_3_OD) δ 27.9. HRMS (ESI^+^): *m*/*z*: Calcd. for C_1__9_H_22_BNO_5_ [M + H]^+^: 356.1663; Found: 356.1656.

##### (4-((2-(Cyclohexylamino)-2-Oxo-1-Phenylethoxy)carbonyl)phenyl)boronic Acid (**5e**)

Following general procedures, the desired compound was synthesized utilizing 4-boronobenzoic acid (93.83 mg, 0.57 mmol), benzaldehyde (0.05 mL, 0.47 mmol), and cyclohexyl isocyanide (0.07 mL, 0.57 mmol), giving compound **5e** as a yellow oil (yield 161.3 mg, 90%). ^1^H NMR (600 MHz, CD_3_OD) δ 8.05 (d, *J* = 8.4 Hz, 2H), 7.82 (br, 1H), 7.58 (d, *J* = 7.2 Hz, 2H), 7.43–7.37 (m, 3H), 6.10 (s, 1H), 3.67–3.63 (m, 1H), 1.91–1.60 (m, 5H), 1.36–1.20 (m, 5H). ^13^C NMR (151 MHz, CD_3_OD) δ 170.3, 167.3, 137.0, 134.8, 129.9, 129.7, 129.6, 128.4, 77.4, 33.5, 33.3, 26.5, 26.0, 26.0. ^11^B NMR (192.5 MHz, CD_3_OD) δ 27.0. HRMS (ESI^+^): *m*/*z*: Calcd. for C_21_H_24_BNO_5_ [M + H]^+^: 382.1820; Found: 382.1823.

##### (4-((1-(4-Bromophenyl)-2-(Tert-Butylamino)-2-Oxoethoxy)carbonyl)phenyl)boronic Acid (**5f**)

Following general procedures, the desired compound was synthesized utilizing 4-boronobenzoic acid (107.60 mg, 0.65 mmol), 4-bromobenzaldehyde (100 mg, 0.54 mmol), and tert-butyl isocyanide (0.07 mL, 0.65 mmol), giving compound 5f as a yellow oil (yield 157.1 mg, 67%). ^1^H NMR (600 MHz, CD_3_OD) δ 8.13 (s, 2H), 8.03–7.84 (m, 2H), 7.71 (s, 2H), 7.61 (q, *J* = 12 Hz, 2H), 6.05 (s, 1H), 1.30 (s, H). ^13^C NMR (151 MHz, CDCl_3_) δ 167.4, 164.9, 134.6, 134.0, 133.6, 132.0, 130.7, 129.0, 128.8, 123.2, 51.9, 28.6. ^11^B NMR (192.5 MHz, CD_3_OD) δ 27.9. HRMS (ESI^+^): *m*/*z*: Calcd. for C_19_H_21_BBrNO_5_ [M + H]^+^: 434.0760; Found: 434.0768.

##### (4-((2-(Tert-Butylamino)-1-(4-Fluorophenyl)-2-Oxoethoxy)carbonyl)phenyl)boronic Acid (**5g**)

Following general procedures, the desired compound was synthesized utilizing 4-boronobenzoic acid (80.20 mg, 0.48 mmol), 4-fluorobenzaldehyde (50.00 mg, 0.40 mmol), and tert-butyl isocyanide (0.05 mL, 0.48 mmol), giving compound **5g** as a yellow oil (yield 128.4 mg, 86%). ^1^H NMR (600 MHz, CD_3_OD) δ 8.04 (d, *J* = 5.4 Hz, 2H), 7.90 (s, 1H), 7.85 (br, 1H), 7.64–7.61 (m, 2H), 7.14 (t, *J* = 9.0 Hz, 2H), 6.07 (s, 1H), 1.31 (s, 9H). ^13^C NMR (151 MHz, CD_3_OD) δ 170.4, 167.3, 164.5 (d, *J* = 247.6 Hz, CF), 135.1, 134.7, 133.5, 130.8, 130.7, 129.8, 116.6, 116.5, 76.8, 52.7, 28.9. ^11^B NMR (192.5 MHz, CD_3_OD) δ 28.1. HRMS (ESI^+^): *m*/*z*: Calcd. for C_19_H_21_BFNO_5_ [M + H]^+^: 374.1561; Found: 374.1569.

##### (4-((2-(Tert-Butylamino)-1-(2-Fluorophenyl)-2-Oxoethoxy)carbonyl)phenyl)boronic Acid (**5h**)

Following general procedures, the desired compound was synthesized utilizing 4-boronobenzoic acid (80.20 mg, 0.48 mmol), 2-fluorobenzaldehyde (50.00 mg, 0.40 mmol), and tert-butyl isocyanide (0.05 mL, 0.48 mmol), giving compound **5h** as a yellow oil (yield 67.2 mg, 45%). ^1^H NMR (600 MHz, CD_3_OD) δ 8.00 (d, *J* = 7.8 Hz, 2H), 7.83 (s, 1H), 7.72 (s, 1H), 7.61 (t, *J* = 7.2 Hz, 1H), 7.39–7.34 (m, 1H), 7.22–7.17 (m, 1H), 7.14–7.10 (m, 1H), 6.41 (s, 1H), 1.29 (s, 1H). ^13^C NMR (151 MHz, CD_3_OD) δ 169.2, 167.0, 162.1 (d, *J* = 249.2 Hz, CF), 134.9, 132.2, 125.6, 124.6, 116.6, 71.4, 52.7, 28.9. ^11^B NMR (192.5 MHz, CD_3_OD) δ 28.3. HRMS (ESI^+^): *m*/*z*: Calcd. for C_19_H_21_BFNO_5_ [M + Na]^+^: 396.1389; Found: 396.1392.

##### (4-((2-(Tert-Butylamino)-2-Oxo-1-(Quinolin-4-yl)ethoxy)carbonyl)phenyl)boronic Acid (**5i**)

Following general procedures, the desired compound was synthesized utilizing 4-boronobenzoic acid (63.34 mg, 0.38 mmol), quinoline-4-carbaldehyde (50.00 mg, 0.32 mmol), and tert-butyl isocyanide (0.43 mL, 0.38 mmol), giving compound **5i** as a yellow oil (yield 59.8 mg, 46%). ^1^H NMR (600 MHz, CD_3_OD) δ 8.89 (s, 1H), 8.37 (d, *J* = 8.4 Hz, 1H), 8.08 (d, *J* = 8.4 Hz, 1H), 8.01 (d, *J* = 7.8 Hz, 2H), 8.79 (t, *J* = 7.8 Hz, 3H), 7.75 (d, *J* = 4.2 Hz, 1H), 7.67 (t, *J* = 7.8 Hz, 1H), 6.86 (s, 1H), 1.32 (s, 9H). ^13^C NMR (151 MHz, CD_3_OD) δ 168.7, 167.0, 151.1, 149.3, 143.9, 134.9, 131.2, 130.2, 129.8, 128.7, 127.6, 125.5, 121.9, 74.3, 52.9, 28.8. ^11^B NMR (192.5 MHz, CD_3_OD) δ 26.9. HRMS (ESI^+^): *m*/*z*: Calcd. for C_22_H_23_BN_2_O_5_ [M + H]^+^: 407.1759; Found: 407.1772.

##### (4-((2-(Cyclohexylamino)-2-Oxo-1-(Quinolin-4-yl)ethoxy)carbonyl)phenyl)boronic Acid (**5j**)

Following general procedures, the desired compound was synthesized utilizing 4-boronobenzoic acid (63.34 mg, 0.38 mmol), quinoline-4-carbaldehyde (50.00 mg, 0.32 mmol), and cyclohexyl isocyanide (0.48 mL, 0.38 mmol), giving compound **5j** as a yellow oil (yield 112.0 mg, 81%). ^1^H NMR (600 MHz, CD_3_OD) δ 8.90 (d, *J* = 4.8 Hz, 1H), 8.41 (d, *J* = 8.4 Hz, 1H), 8.10 (d, *J* = 8.4 Hz, 1H), 8.04 (d, *J* = 8.4 Hz, 2H), 7.83–7.77 (m, 4H), 7.69 (t, *J* = 7.8 Hz, 1H), 6.90 (s, 1H), 3.70–3.67 (m, 1H), 1.89–1.57 (m, 5H), 1.35–1.19 (m, 5H). ^13^C NMR (151 MHz, CD_3_OD) δ 167.9, 166.2, 150.4, 148.5, 142.9, 134.3, 130.5, 129.4, 129.1, 128.0, 126.7, 124.8, 121.2, 73.5, 32.7, 32.6, 25.8, 25.4. ^11^B NMR (192.5 MHz, CD_3_OD) δ 27.2. HRMS (ESI^+^): *m*/*z*: Calcd. for C_24_H_25_BN_2_O_5_ [M + H]^+^: 433.1929; Found: 433.1908.

##### (4-((2-(Cyclohexylamino)-1-(4-Fluorophenyl)-2-Oxoethoxy)carbonyl)phenyl)boronic Acid (**5k**)

Following general procedures, the desired compound was synthesized utilizing 4-boronobenzoic acid (63.34 mg, 0.38 mmol), 4-fluorobenzaldehyde (0.03 mL, 0.32 mmol), and cyclohexyl isocyanide (0.48 mL, 0.38 mmol), giving compound **5k** as a yellow oil (yield 104.7 mg, 82%). ^1^H NMR (600 MHz, CD_3_OD) δ 8.05 (s, 2H), 7.85 (br, 1H), 7.72 (br, 1H), 7.63–7.61 (m, 2H), 7.14 (t, *J* = 9.0 Hz, 2H), 6.08 (s, 1H), 3.66–3.62 (m, 1H), 1.91–1.60 (m, 5H), 1.35–1.17 (m, 5H). ^13^C NMR (151 MHz, CD_3_OD) δ 170.2, 167.2, 164.6 (d, *J* = 246.1 Hz, CF), 135.1, 134.6, 133.3, 130.8, 130.7, 129.9, 116.7, 116.5, 76.8, 33.6, 33.5, 26.7, 26.2. ^11^B NMR (192.5 MHz, CD_3_OD) δ 11.5. HRMS (ESI^+^): *m*/*z*: Calcd. for C_21_H_23_BFNO_5_ [M − H]^−^: 398.1581; Found: 398.1576. 

##### (4-((2-(Tert-Butylamino)-2-Oxoethoxy)carbonyl)-2-Nitrophenyl)boronic Acid (**5l**)

Following general procedures, the desired compound was synthesized utilizing 4-borono-3-nitrobenzoic acid (421.9 mg, 2.00 mmol), formaldehyde (0.16 mL, 1.67 mmol), and tert-butyl isocyanide (0.23 mL, 2.00 mmol), giving compound **5l** as a yellow oil (yield 346.4 mg, 64%). ^1^H NMR (600 MHz, CD_3_OD) δ 8.85 (s, 1H), 8.50–8.44 (m, 1H), 7.78 (t, *J* = 7.8 Hz, 1H), 4.78 (s, 2H), 1.38 (s, 9H). ^13^C NMR (151 MHz, CD_3_OD) δ 168.5, 165.4, 149.7, 136.5, 132.7, 131.2, 128.7, 125.4, 64.6, 52.4, 28.9. HRMS (ESI^+^): *m*/*z*: Calcd. for C_13_H_17_BN_2_O_5_ [M + H]^+^: 325.1202; Found: 325.1122.

### 3.2. Cell Line and Cell Culture

HEK293 cells and human embryonic kidney cells were obtained from the American Type Culture Collection. The cells were maintained in Dulbecco’s modified Eagle’s medium supplemented with 10% FBS, 100 U/mL penicillin and 100 mg/mL streptomycin at 37 °C under 5% CO_2_.

### 3.3. Toxicity Analysis

HEK293 cells were plated at 70% density in 24-well plates at 37 °C. All compounds were first dissolved in 100% DMSO to give 1 mM as a stock for the following dilutions. The cells were treated with all compounds except **4e**, **5a**, **5h**, **5i**, **5l** (for their poor solubility issues) at the dose of 1, 10 and 50 µM for 48h. Further, the cell viability was measured by the 3-(4,5-Dimethyl-2-thiazolyl)-2,5-diphenyl-2H-tetrazolium bromide (MTT) assay. After incubating for 1h, the reaction product was read at 570 nm using a Synergy HT microplate reader (BioTek Instruments, Winooski, VT, USA).

### 3.4. Cellular Bio-Distribution

HEK293 cells were treated with compound **4a**, **5b**, **5f**, **5g** and **5k** (1 mM or 2 mM) for 2h. Then, cells were added DAHMI (2 mM) for another 1h. The complex of DAHMI-Boronic acid compound has an excitation/emission at ~358/461 nm. The distributions of boronic acid compounds were observed by fluorescence microscopy using an Olympus CX Microscope.

## 4. Conclusions

In conclusion, we have developed a boronic acid protection-free microwave-assisted condition to construct two series of depsipeptide boronic acids. The mechanistic study using ^11^B NMR spectrum and theoretical calculations suggested that the location of the boronic acid group plays a significant role in influencing the overall yields. This reported strategy enables the facile relocation of the boronic acid within the molecular framework. Furthermore, the synthesized boronic acids could be subjected to linker-free tagging experiments, where their bio-distribution profiles can be directly assessed.

## Data Availability

Not applicable.

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
