# Peer review of "Protection-Free Strategy for the Synthesis of Boro-Depsipeptides in Aqueous Media under Microwave-Assisted Conditions"

_molecules, 2022, doi:10.3390/molecules27072325_

Round 1

Reviewer 1 Report

First of all, reviewer takes long time to compare with previous version, which part did authors revise.

It is not sense  no descriptions which part did revise for new submissions in cover letter etc.

Reviewer cannot recognize the improvements in new submission,  so judgement did not change at this stage.  

It is not efficient descriptions for biological activities to reproduce by anyone else. (Why did descriptions of DMSO In S67 and S68 ? DMSO cannot be found in 3.2~3.4 in the text body.)

Author Response

Please see the attached pdf file

Reviewer 2 Report

The authors considered most of the comments and suggestions of reviewers and made the appropriate changes, so their work is enriching and can be published in molecules.

Reviewer 3 Report

The authors have developed the efficient and practical boron-containing depsipeptides in aqueous conditions using Psserini three-component reaction with microwave-irradiation. Various boro-depsipeptides were synthesized in moderate to high yields. The authors also revealed that the location of the boronic acid group in the starting material affect the reactivity of the Psserini reaction. Theoretical and 11B NMR analyses are discussed and quite reasonable. The cellular uptake characters of synthesized boro-depsipeptides are investigated to indicate the utility of the products.

On the other hand, the purity of some compounds needs to be improved.

Therefore, this reviewer suggests that the manuscript can be accepted in the Molecules after the appropriate revisions outlined below.

Comments

  1. The peak height of 1H and/or 13C NMR of the compounds 4b, 4d, 4e, 4g, 5a-c, 5f, 5i, 5j, and 5l were too low to characterize and judge. If the solubility of the compounds in CD3OD is an issue, DMSO-d6 should be used.
  2. 1H NMR peak of 4a and 5a-c were not sharp.

3. 4d, 5e-h, and 5k is not enough pure to characterize.

Round 2

Reviewer 1 Report

One of the topics of the manuscript might be difference  of "cellular bio-distribution" between 5f, 5g  and 5k. 

Identifications of these compounds in SI are quite un-reliable. 1) There are over 4 peaks of carbonyl regions in 13C-NMR for 5g and 5k (2 peaks for 5f). 2) Authors presented 13C NMR of 5f was measured in CD3OD. But the solvent signals of this chart might be 77.0 ppm. 3)No descriptions for C-F coupling were presented in all of the text. 4) Integrations of cyclohexane moiety for 5k was calculated over 21H in H-NMR spectrum. 

These results the purities of the synthetic compounds are quite unreliable.

Furthermore, their synthetic routes synthesized racemates for compound 5 series. If the proportions of racemate might be influenced for their biological activity, the results in Table 5 have many factors to discussion. 

Author Response

Enclosed is a point-to-point response to the reviewer’s comments regarding our re-submission.

We hope these responses could provide sufficient information that Molecules would re-consider to publish our work.

Thank you for your consideration. I look forward to hearing from you.

Sincerely,

This manuscript is a resubmission of an earlier submission. The following is a list of the peer review reports and author responses from that submission.

Round 1

Reviewer 1 Report

The manuscript titled " Protection-Free Strategy for the Synthesis of Boro-depsipeptides in Aqueous under Microwave-assisted Conditions" reports the synthesis of 17 boron-containing depsipeptides via microwave-assisted in an aqueous media. The authors have demonstrated that the preparation of boron-containing depsipeptides was achieved by a one-pot three-component reaction from Carboxylic acid, aldehyde and isocianyde in aqueous media under microwave irradiation. The description of these syntheses and the characterization of these compounds are well explained. Meanwhile, a few synthesized boron-containing compounds were tested to investigate the relationship between their structures and their level of cellular uptake of HEK293 cells using the linkerless DAHMI fluorescent labeling approach.

Overall the manuscript is rich and interesting; and the paper structure is well-knit and suitable for publication in the journal, after minor revisions. The comments are listed as the following points:

  1. Title, " Protection-Free Strategy for the Synthesis of Boro-depsipeptides in Aqueous under Microwave-assisted Conditions" should be " Protection-Free Strategy for the Synthesis of Boro-depsipeptides in Aqueous Media under Microwave-assisted Conditions"
  2. In section 2.1, the authors should explain why the yield decreases with increasing reaction time.
  3. In Table 1 and all text, "µW" should be "MW".
  4. In Table 2, the authors should explain the choice of temperature 25°C for entry 1 (under stirring), and temperatures 65, 85 and 95°C for entries 2-4 (under MW). Why did they not use the optimal condition obtained for entry 5, Table 1 (under MW at 45 °C for 2.5 hours).
  5. Page 8, line 178, “table 5” should b “Table 5”.
  6. In section 2.3. In Vitro Studies, the authors should justify the choice of the tested compounds (4a, 5b, 5f, 5g and 5k) and why not the other compounds.
  7. I believe that the results of the toxicity analysis were not discussed in the results and discussion section.
  8. Page 10, line 203 and Page 11, line 277, “D.I” should be “Deionised”
  9. Synthesis and characterization of compounds, there are no HRMS data of compounds 4e, 4g and 5h.
  10. Supporting information, there are no HRMS spectra of the synthesized compounds.

Author Response

Dear Reviewer 1,

Enclosed is a point-to-point response to your comments regarding our revised submission. 

We hope these responses could provide sufficient information that Molecules would re-consider to publish our work.

Thank you.

Reviewer 2 Report

Pan and co-worker report in this manuscript the Passerini three-component synthesis of boronic acid-containing depsipetdies with their cellular bio-distribution study. The boron effect in the substrates for Passerini reaction is uniquely rationalized by theoretical calculations and control experiments for the first time in this study. The cellular bio-distribution study based on the DAHMI method is new in evaluating in vitro profile of boron-containing depsipetdies. Considering these new points, this work is provisionally acceptable in Molecules journal, but I cannot recommend its publication in the present style due to the lack of sufficient introduction of background and related studies. The similar multicomponent reaction for phenyl boronic acid was previously suggested by other researchers, which provided several numbers of depsipetdies with free boronic acid moiety (Eur. J. Org. Chem. 2019, 6132 unreferenced and ACS Omega 20183, 7783). The authors cite the latter literature as a reference for their biological evaluations, while this ACS Omega work surely included the Passerini three-component synthesis of boronic acid-containing depsipetdies. I would understand that the Pan group is an early contributor for synthesis of boron-containing peptides and development of multicomponent reactions, but depsipetdies with free boronic acid moiety were first reported by other researchers. Thus, the authors should provide more appropriate introduction and citations for synthetic study of depsipetdies with free boronic acids, otherwise the manuscript is not suitable for publication as an article in a reputed high impact factor journal, Molecules.

Author Response

Dear Reviewer 2,

Enclosed is a point-to-point response to your comments regarding our revised submission. 

We hope these responses could provide sufficient information that Molecules would re-consider to publish our work.

Thank you.

Reviewer 3 Report

Synthesis of phenylboro-depsipeptides with microwave was described.

The manuscript was not well organized, reviewer cannot agree to publish the manuscript to Molecules

1) Figure 1, There are no phenylboronic acid contained compounds in the figure. Authors have to demonstrate to synthesis several compounds in the figures.

2) Table 1, 2-1) Microwave irradiations in organic solvents have to demonstrated.  2-2) The decomposition of 4a  with microwave at 45 °C  has to demonstrated. 2-3) What kinds of side reaction occurred inter settings?

3) Scheme 1; All reactions was setting with tert-Butyl Isocyanide. There is no necessary to describe R3.

4) Table 2; 4-1) It is essential to describe the reaction at 85 °C without microwave irradiation. 4-2) The decomposed products have to analyzed at 95 °C

5) Scheme 2; Different substitutions on phenylboronic acids are listed only 2 examples. It is very difficult to distinguish substitution effects on the components. 

6) Table 5; 6-1) The structure of 4a was different between scheme 1 and table 5. Reviewer cannot rely their biological data., 6-2) No synthetic descriptions for 5k in the manuscript (Scheme 2). 6-3) All synthetic compounds are conducted to cellular bio-distribution to compare structure-activity relationships.

Author Response

Dear Reviewer 3,

Enclosed is a point-to-point response to your comments regarding our revised submission. 

We hope these responses could provide sufficient information that Molecules would re-consider to publish our work.

Thank you.

Round 2

Reviewer 2 Report

I would now recommend acceptance of the present manuscript in Molecules journal. In this revised version, the authors have made suitable revisions according to the reviewers’ comments. In my view, this work provides all the necessity experimental results for understanding the target reactions. The early and related studies of multicomponent reaction with free boronic acids and its application to depsipetdie synthesis are suitably introduced in the contents. With such proper improvements, the present manuscript has become suitable for publication as a Special Issue article in Molecules.

Reviewer 3 Report

1) Figure 1: Author added L-BPA in the figure. Is it available to synthesis L-BPA with their presentation method? Reviewer pointed out no compounds listed Figure 1 can be synthesized by their presentation method. 

2) No microwave effects are described in the text body. for example lane 84-108: Increasing the reaction temperature from room temperature to 45°C with H2O significantly improved the yield from 5% to 68% (en- 108 try 7).

They compared 25° C without and 45 °C with microwave irradiations. It is difficult to compare only temperature effect.

3) It is very difficult to read the sentence from 139 to lead Table 2. Ie is not essential chemical yield of 4a in Table 1. 

The second series of analogues were constructed using boron-containing benzoic acids as one of the building blocks. The optimization process is summarized in Table 2. By employing the optimized condition from series 1 (Table 1, entry 7), the desired product 4a was obtained in 68%. Different to what we’ve observed from Table 1, it was found that the yield would be significantly improved if the reaction temperature were raised from ambient temperature to 85 °C (Table 2, entry 1, 4-6). 

4)Figure 3, If the complex of (HO)2B-benzaldehyde + benzoic acid afforded tetra-coordinate boron, pKa of acid components sounds important for the reaction in Scheme 1. Please compare the roles of acid components , especially 4g, (why did not the other acid compornents inhibit the reactions?)    

5) Table 5, All tested compound has cytotoxicity at above 50 microM in HEK-293 cells. Did BNCT at 1or 2 mM perform to detect dead cell? BNCT analysis at 50 microM will be preferred.